# Effectiveness of Pelvic Floor Muscle Training on Quality of Life in Women with Urinary Incontinence: A Systematic Review and Meta-Analysis

**DOI:** 10.3390/medicina59061004

**Published:** 2023-05-23

**Authors:** César Adrián Curillo-Aguirre, Enrique Gea-Izquierdo

**Affiliations:** Faculty of Medicine, Pontifical Catholic University of Ecuador, Quito 170143, Ecuador

**Keywords:** urinary incontinence, pelvic floor muscle training, quality of life

## Abstract

*Background and Objectives:* Urinary incontinence (UI) is a condition that is more common in women than men and has an increasing prevalence with age. It provides a range of psychological and physical burdens that negatively affect the patient’s quality of life (QoL). However, the economic burden for the healthcare system is being augmented due to the increasing life expectancy of the population. This article aims to identify the effectiveness of pelvic floor muscle training (PFMT) on the QoL in women with UI. *Materials and Methods*: A systematic review and meta-analysis were conducted in the PubMed, EMBASE, ProQuest medicine, Cochrane Library, and Google Scholar databases. The terms selected according to components of PICOS were women with urinary incontinence, pelvic floor muscle training, watchful or other types of therapies, quality of life, randomized controlled trials, and interventional or observational studies. The articles included were those published between November 2018 and November 2022. Ten articles were found for the systematic review and eight for the meta-analysis. *Results*: The QoL moderately increased when PFMT was used on women with UI, the results indicating an overall small effect on the QoL across the controlled studies and a moderate effect on the QoL across the one-group pre-post-studies. *Conclusions*: Specific QoL domains, such as social activities and general health, also demonstrated benefits from PFMT interventions. This study confirmed the effectiveness of PFMT on the QoL in women with UI, mainly for patients with stress urinary incontinence.

## 1. Introduction

As stated by the International Continence Society (ICS), urinary incontinence (UI) is defined as any leak or involuntary loss of urine that implies a social and hygienic problem [1]. UI affects both sexes. However, women are more affected than men [2]. The prevalence of UI worldwide is estimated to be around 17% to 30% in women older than 20 years and 38% to 50% in women older than 60 years [3,4]. The incidence of UI increases with age until it becomes a geriatric syndrome [5]. It is estimated that up to 77% of women residing in nursing homes have UI [6].

According to the World Health Organization (WHO), between 2015 and 2050, the percentage of the planet’s inhabitants over 60 years of age will almost double, from 12% to 22% [7]. As a result of the population’s increasing age, the healthcare system must find ways to guarantee adequate medical and social assistance to the specific needs of this population. Patients with UI have higher rates of hospitalization, urinary tract infections, pressure ulcers, admission to assisted living homes [8], lower work productivity, social isolation, and higher rates of depression [9], which results in a significant reduction on their quality of life (QoL) [10].

Multiple risk factors have been associated with UI, the most common being overweight, obesity, female sex, vaginal delivery, multiparity, menopause, advanced age, and previous genitourinary and abdominal surgeries [11]. On the other hand, certain risk factors remain controversial, such as a sedentary lifestyle, smoking, caffeine intake, and high-impact sport practice [12]. However, most researchers agree that the main risk factor is an increase in intra-abdominal pressure that is most seen in overweight and obese patients [13].

UI can be classified as stress, urgency, mixed, overflow, or functional UI. Stress urinary incontinence (SUI) occurs with increases in intra-abdominal pressure such as laughter, coughing, sneezing, exertion, or physical activity, due to weakening of the urethral sphincter [14]. Urge urinary incontinence (UUI) may be preceded or accompanied by a sensation of urinary urgency due to detrusor overactivity. It can also be triggered by the sound of running water, drinking cold beverages, or exposure to cold temperatures [15]. Mixed urinary incontinence (MUI) is caused by a combination of urgency and stress urinary incontinence [16].

It is important to keep in mind that the pelvic floor musculature (PFM) is a very complex structure, playing an essential role in the micturition process by providing stability and support to the muscles, aponeurotic elements, and ligaments that participate in micturition and fecal dynamics [17]. The perineal membrane is currently described as the most important structure in urinary continence, since it is located in the urogenital triangle and is responsible for stabilizing and compressing the external urethral sphincter [18]. This set of structures is often called the urogenital diaphragm, which is a continuation of the inferior fascia of the levator ani muscle, the main structure of the pelvic diaphragm [19].

Defining the underlying cause of UI is essential to guide an appropriate treatment. Multiple techniques are available for the treatment of UI, and these should be adjusted based on the patient rather than the type of incontinence. Treatment options include a conservative, pharmacological, or surgical approach [15]. This research focuses mainly on the conservative treatment of UI, emphasizing pelvic floor muscle training (PFMT).

The conservative approach is considered the first choice of treatment due to its accessibility, cost-effectiveness, and limited risks and side effects compared to more invasive treatments. Conservative management consists of lifestyle changes and exercises that strengthen the PFM. The most effective treatment is the PFMT paired with lifestyle changes such as adequate fluid intake, scheduled urination, decreased carbonated and caffeinated beverage intake, smoking cessation, moderate physical activity, avoiding tight clothing, and weight loss if necessary [20].

PFM strengthening exercises were described 72 years ago by the American gynecologist Arnold Kegel in the 1950s, who described that muscular activity causes muscles to lose four times less mass compared to muscles that remain inactive [21]. Several studies have reported that training regimens that strengthen the pelvic floor muscles have effectiveness rates from 29% to 59%. These should be practiced as intensely and frequently [22] as possible, provided by qualified personnel [23], and maintained for at least 3 months before reviewing its effectiveness [24].

Training effectiveness changes from patient to patient. A systematic review found that, in SUI, 58.8% of patients achieved a significant improvement after 12 months of supervised PFMT, reached a 17% improvement after 12 months in UUI, and, in MUI, 28% of patients had improved symptoms and QoL after 6 months [23].

UI considerably affects people’s QoL, causing discomfort and shame in those who suffer from it, as well as having enormous psychological repercussions on the patient and their family. Women often limit their social interactions or completely isolate themselves to avoid involuntary urinary leakage at social settings or workplaces and thus have higher rates of depression and low self-esteem due to this social and hygienic issue.

This investigation aims to indicate the effectiveness of pelvic floor strengthening exercises and their impact on the QoL. Furthermore, the risk factors associated with UI and which type of UI is associated with the greatest improvement in the QoL using PFMT will be assessed. This analysis will provide an updated and comparative theoretical–methodological approach to the treatment of UI.

## 2. Materials and Methods

This systematic review and meta-analysis were developed following the PRISMA 2020 checklist [25].

Given the nature of the study, it was not necessary to obtain informed consent from patients.

### 2.1. Eligibility Criteria

This review was conducted by the investigators with all the titles and abstracts retrieved during the initial search. Articles for assessment were included separately by the investigators. For the syntheses, studies were grouped depending on their design.

#### 2.1.1. Inclusion Criteria

Articles that mentioned any type of exercise aimed at training and strengthening the pelvic floor as a therapeutic measure for the treatment of urgency, stress, and mixed UI. Articles were selected in English and with a publication date between November 2018 and November 2022, chosen depending on their free and unrestricted accessibility.

#### 2.1.2. Exclusion Criteria

Articles that studied UI due to nerve injury or a neurogenic bladder; articles that included in their studies pregnant women, any degree of pelvic organ prolapse, a history of neurological disease or epilepsy, the use of a surgical or pharmacological treatment; and articles that did not meet the inclusion criteria described above were excluded from this study.

### 2.2. Information Sources

A systematic literature review was completed through electronic sources such as PubMed, EMBASE, ProQuest medicine, the Cochrane Library, and Google Scholar, which contain original studies, approved by high-impact journals in the areas of medicine and epidemiology, published in the last 5 years between November 2018 and November 2022. Databases from inception until 30 December 2022. Any inconsistencies were resolved by consensus between the authors.

### 2.3. Search Strategy

In PubMed, the following search strategy was used: (urinary incontinence) OR (urinary incontinence, stress/prevention, control OR urinary incontinence, and stress/rehabilitation) OR (urinary incontinence, urge/prevention, and control) OR (pelvic floor exercise) OR (pelvic floor muscle training OR pelvic floor disorders/rehabilitation) AND (quality of life).

MeSH terms selected according to components of PICOS. P (Population): women with urinary incontinence, I (Intervention): pelvic floor muscle training, C (Comparison group): watchful or other types of therapies, O (Outcome): quality of life, S (Observational): randomized controlled trials and interventional or observational studies.

### 2.4. Selection Process

This review was executed based on the recommendations of the Preferred Reporting Items for Systematic Reviews and Meta-Analyses (PRISMA) guidelines. Original research articles that provided mean differences and standard derivations of the effectiveness of PFMT in women with UI were eligible for this meta-analysis. Both randomized controlled trials and observational studies were accepted. The variables of the articles included were risk factors, incidences, and the prevalence of UI. The main variables that were considered for this study were age, sex, body mass index, parity, type of childbirth delivery, menopause, pelvic surgeries, smoking, alcohol and caffeine intake, PFMT, QoL, urinary incontinence, SUI, UUI, MUI, urinary tract infection, frequency, and amount of urine loss. Articles found to be duplicates were excluded from the study at this stage, as shown in Figure 1.

### 2.5. Data Collection Process

Data extraction was performed by the author using a standardized form. The following data were extracted: authors, year of publication, study design, sample size, stratification, country, age, sex, UI diagnosis, type of UI, and type of intervention.

Quality checks were also performed to compare the abstracted data with the original articles. In the article search, a total of 737 results were obtained. One hundred and ninety-five articles continued to be relevant to this review, of which one hundred and eighty-five were excluded for the following reasons: inappropriate outcome for the purpose of this review, interventions using pharmacological or surgical treatments, other types of UI, pregnant women, studies with animals, and those that did not meet the inclusion criteria. At last, 10 relevant articles were found for the systematic review and 8 articles for the meta-analysis.

### 2.6. Data Items

For each eligible study, two investigators extracted the name of the first author, year of publication, setting, sample size, mean age of the population, diagnostic tool used for the QoL, and the severity of the UI.

### 2.7. Outcomes

The primary outcomes were considered the mean values and the correspondent standard deviations (SDs) of the validated tools of the QoL, comparing the values of participants with UI and the controls. If the data were reported in other ways (median and interquartile ranges), they were transformed into means and SD.

### 2.8. Study Risk of Bias Assessment

Two independent authors made assessments of the studies’ quality. Evaluating the risk of bias of all the selected articles was done using a modified Cochrane tool for assessing the risk of bias, the criteria outlined in the Cochrane Handbook for Systematic Reviews of Interventions. The risk of bias was classified as high risk, low risk, or unclear risk for each of the following six domains: random sequence generation (selection bias), allocation concealment (selection bias), blinding of participants and personnel (performance bias), blinding of the outcome assessment (detection bias), incomplete outcome data (attrition bias), and selective outcome reporting (reporting bias).

### 2.9. Effect Measures

We pooled the results of all the studies, regardless of the type of exercise intervention. The main measure of the effect for the meta-analysis was the means difference. We estimated the differences between means by the standardized difference in means (SMD). According to the studies’ differences, random effects models of analyses were used. Most studies reported preintervention and postintervention mean differences (MD). Differences between the intervention and control groups were calculated and assessed.

### 2.10. Synthesis Methods

A database was prepared with the elements mentioned in the previous section. In addition, the standard mean differences and their confidence intervals were reported with the Review manager program (RevMan 5.4.1) for Windows. Heterogeneity was evaluated by the I-squared index (I2 index) and the chi-square test, with similar effects. The level of statistical significance was set at *p* < 0.05. Subsequently, forest plot graphics were conducted to illustrate the statistical data, and tables and graphs were generated to establish the comparative analyses.

A discussion of the information collected in relation to the prevalence of UI risk factors was established, as well as the pathophysiology and presentation of the diseases found in the different investigations analyzed, as this allowed establishing similarities and differences to support our conclusions.

### 2.11. Meta-Analysis

Of the eight studies extracted related to UI risk factors, it was possible to specify that the patients who improved were slightly older than the controls, and there were more patients with SUI.

In the meta-analysis, a random Mantel–Haenszel model from the RevMan 5.4.1 program was used to assess the clinical presentation of the study group.

In addition, a *p*-value (significance level < 0.05) analysis was used to validate the hypothesis framed in the previous protocol, which established that the multiparity risk factor affects women with UI more.

## 3. Results

### 3.1. Study Selection

The systematic review included 10 articles with a sample of 1648 women from nine countries. In general, the sample sizes were small to moderate in size, with follow-ups generally of less than 24 weeks, and many were at moderate risk of bias. There was considerable variation in the type of intervention and duration of the intervention, study populations, and outcome measures. Finally, 10 articles were eligible to enter in our review.

This meta-analysis analyzed eight studies for the general QoL, of which seven studies evaluated PFMT. Two interventional studies, one assessing Iyengar yoga and the other evaluating modified Pilates for the strengthening of pelvic floor muscles, were excluded due to insufficient statistical data and the nature of their study designs.

Decisions for the inclusion of articles were made by agreement between the two reviewers. Table 1 shows an outline of the ten included studies. Articles were arranged by publication date, from most recent to oldest. The duration of the intervention ranged from a total of 6 to 24 weeks. For the comparison group, the control subjects in most of the studies received PFMT without supervision or just information about the exercises. Two studies did not have a control group due to their designs. All studies included only women in their samples, and most of the articles evaluated the QoL of the participants through questionnaires validated for this purpose at the beginning and after the intervention.

Two studies used a different type of exercise to strengthen the pelvic floor. The first study described Iyengar yoga therapy, a regimen that consisted of a two-day course of supervised group yoga and a one-day course of yoga at home weekly for eight weeks, each session lasting for 75 min [26]. The investigators concluded that yoga improved pelvic muscle strength and the QoL. The second study combined modified Pilates (MP) with standard physiotherapy care. The regimen consisted of standard physiotherapy sessions before the MP classes, where women attended six one-hour group classes of six to eight people run at one-week intervals by a specialist in Pilates for 24 months [35]. The results revealed multiple benefits, such as improved self-esteem (*p* = 0.032) and decreased social embarrassment (*p* = 0.026), while women with higher symptom severity showed improvement in their personal relationships (*p* = 0.017). A qualitative analysis supported these findings and indicated that MP classes could positively influence attitudes towards exercise, diet, and wellbeing.

Only one study compared the effectiveness of PFMT between parous and nulliparous women, where the authors claimed that UI is more common among parous women and presents with more severe symptoms and lower health-related QoL. However, PFMT seems to be effective in improving the symptoms of UI and health-related QoL among both parous and nulliparous women [31].

Eight studies were included to assess the QoL. Figure 2 shows a forest plot of a meta-analysis where the pooled MD of the QoL between the experimental and control groups was −3.19 with a CI [−5.99 to −0.40] and *p* = 0.03. A meta-analysis of these trials revealed that PFMT can mildly improve the QoL.

The heterogeneity of the included trials was high (I2 = 97%, *p* < 0.00001). A regression meta-analysis should be performed to evaluate the sources of heterogeneity. However, the scant number of studies included in this meta-analysis was not sufficient to run a statistical subgroup analysis. To obtain the forest plot, random effects were used due to the different types of interventions used in the studies.

### 3.2. Study Characteristics

The characteristics of the considered studies are shown in Table 2.

The following questionnaires were used in the QoL assessment: the International Consultation on Incontinence Questionnaire-Urinary Incontinence Short Form (ICIQ-UI SF), ICIQ Lower Urinary Tract Symptoms Quality of Life Module (ICIQ-LUTSQoL), King’s Health Questionnaire (KHQ), Incontinence Quality of Life Questionnaire (I-QoL), and Five-Dimensional Euro QoL Questionnaire (EQ5D). The psychometric qualities, accuracy, and credibility of all the questionnaires were confirmed clinically. The most used questionnaire was the ICIQ-UI SF; it was mentioned in 7 out of the 10 included studies.

The ICIQ-UI SF scores range from 0 to 21 and are the weighted sum of three items addressing urinary incontinence frequency (“How often do you leak urine?” 0 = never to 5 = all the time), leakage quantity (“How much urine do you usually leak?” 0 = none to 6 = a large amount), and interference with everyday life (0 = not at all to 10 = a great deal). Higher scores reflect a greater severity [36]. The Kings Health Questionnaire (KHQ) is a patient self-administered, self-reporting tool composed of three parts of 21 items that identifies features such as limitations in performing daily activities, social relationships, emotions, sleep, and energy. It is a validated questionnaire recommended by the European Clinical Practice Guidelines to assess the QoL of patients with UI [37].

Five studies were included to assess the QoL before and after their interventions. Figure 3 shows a forest plot with data from the experimental groups before and after their interventions. The MD of the ICIQ-UI SF questionnaire was 3.92, with a CI [2.97–4.86] and *p* = 0.00001. A meta-analysis of these trials revealed that the ICIQ-UI SF improved approximately three points from the baseline in most studies, thus showing that the QoL of all the women of the experimental groups improved when performing supervised PFMT.

The heterogeneity of the included trials was low (I2 = 27%, *p* < 0.00001). To obtain the forest plot, random effects were used due to the different types of interventions used in the studies.

### 3.3. Risk of Bias in Studies

The risk of bias graph and the risk of bias summary are shown in Figure 4 and Figure 5. One study had an unclear risk of selection bias due to insufficient information provided on random sequence generation or allocation concealment [34]. Four studies reported adequate allocation concealment using sequentially numbered, sealed opaque envelopes [27,30,32,34].

Three studies reported an unclear risk of performance bias [29,30,35], two studies reported a high risk of performance bias [32,33], and only two study reported a low risk of performance bias [27,28]. Since PFMT is an exercise-based therapy, it was not feasible to consider blinding the women or their health providers, as stated by four studies that showed an unclear risk in relation to blinding of the outcome assessment (detection bias) [27,28,32,34].

Seven studies had low risks of selection bias due to the validated tools they used to generate a random sequence between groups [27,28,29,30,32,33,35]. As shown in Figure 4, the random sequence generation and the allocation concealment showed the least risk of bias among all the studies included.

Regarding the assessment of incomplete outcome data, two studies were rated as low risk [29,35]. Five studies had an unclear risk for attrition bias [27,28,30,33,34] due to the unclear reporting of results and uncertainty towards the outcomes and measurements. A low risk of reporting bias was given for four studies [27,28,32,33].

## 4. Discussion

In this systematic review and meta-analysis, ten studies were used; they showed the data of 1648 women from nine different countries from 2018 to 2022. We found that the incidence of UI has been increasing over the years and that there has also been a statistically significant increase in the QoL when pelvic floor muscle exercises have been used to treat UI women.

In the results, we described that PFMT was significantly associated with an increase in the QoL. Multiple studies included in this review reported that UI impacts a wide range of life domains, as shown by Pizzol et al. [38] in a meta-analysis with over 23 articles and a total of 24,983 participants, where UI was significantly associated with a poor QoL, in different areas; among the most important were physical health, emotional health, energy, emotions, social activities, and general health.

There are several validated questionnaires available to measure the QoL. Three questionnaires (EQ-5D-3L, ICIQ-UI SF, and POP-SS) were compared in a study conducted by Fenocchi et al. [39] that used data from the Optimal Pelvic floor muscle training for Adherence Long-term (OPAL) trial. The ICIQ-UI SF questionnaire proved to be effective in identifying a decline in the quality of life, especially in severe cases of UI. In addition, another study by Mangir et al. [40] attributed a positive predictive value (PPV) of 90%, which proved to be useful in applying to the general population in primary care. The ICIQ-UI SF questionnaire was identified as an optimal tool to screen, diagnose, and evaluate the degree of urinary incontinence. Additionally, it is widely used as a tool to estimate the quality of life before, during, and after interventions, as shown in most studies used in this research.

The benefit of PFMT between parous and nulliparous women was compared by Szatmári et al. [31] in 67 patients. The participants were assessed before and after 10 weeks of pelvic floor exercises using the ICIQ-UI SF questionnaire, searching to identify the QoL and symptom severity. Before the intervention, parous women (75%) were two times more likely to report urinary incontinence than nulliparous women (37.5%), and they had significantly lower pelvic floor muscle strength (*p* = 0.001), pelvic floor muscle endurance (*p* = 0.001), and more severe symptoms related to urinary incontinence (*p* = 0.009). Additionally, their findings suggested that pelvic floor muscle training is an effective conservative treatment option for improving the symptoms of UI and health-related QoL among parous and nulliparous women.

Multiple risk factors besides the number of vaginal deliveries, such as residence, physical activity, menopause, and smoking, were identified by Ptak et al. [33]. They assured that more than three vaginal deliveries contribute to the insufficiency of the pelvic floor muscles. This hypothesis has been supported by numerous studies describing the connection between urinary incontinence and the number of deliveries. Lassere et al. [41] reported an odds ratio (OR) of 4.1 for women who had delivered more than three times and an OR of 3.0 for women who had delivered only twice. Likewise, Ozdemir et al. [42] studied 233 women with urinary incontinence and their QoL, as well as their PFM strength; in their results, they published that higher numbers of vaginal deliveries resulted in statistically significant decreased QoL and poor PFM function.

In China, Pang et al. [43] studied a group of 24,985 women between May 2014 and March 2016. They found that the risk ratio (RR) for spontaneous vaginal delivery was 2.12 and 3.30 for instrumental delivery. Other risk factors included a high body mass index (BMI) (overweight RR 1.52 and obesity RR 1.67), cigarette smoking (RR 1.54), chronic cough (RR 1.44), diabetes (RR 1.33), and older age (50–59 years RR 1.49 and 60–69 years RR 1.61).

Considering the various protocols and training regimens aimed at strengthening the pelvic floor muscles, we identified that, for the most part, a 12-week regimen is an adequate therapy duration. Al Belushi et al. [30] intervened with a well-structured 12-week training protocol that consisted of two phases that involved endurance and speed training. The endurance training (tonic contractions) of the PFMs consisted of a slow velocity close to the maximum contractions for 3 to 10 s (the period of contraction was increased by 1 s per week to a maximum of 10 s). Speed training (phasic contractions) involved fast contractions of moderate strength for 2 s followed by relaxation for 2 s. The goal was to have five home sessions of both slow and fast contractions per day in the supine, sitting, and standing positions. In their results, they concluded that the home-based PFMT was effective in reducing the severity of the symptoms and improving the QOL in women with SUI.

The NICE guidelines for UI recommend supervised pelvic floor muscle training for at least 3 months (12 weeks). If pelvic floor muscle contractions are confirmed, women are usually supervised through three appointments during this time [44].

In a meta-analysis, Hadizadeh-Talasaz et al. [45] sought to identify the effect of pelvic floor exercise on female sexual function and quality of life in the postpartum period, where they found that pelvic floor muscle training in primi- or multi-parous women can boost sexual function and quality of life in postpartum women. Many mechanisms explain why using pelvic floor muscle exercises can improve sexual function. Pelvic floor exercises strengthen the levator-ani muscle through muscular hypertrophy. Stronger levator ani muscle enhances the support and lessens the burden imposed on the ligament. In addition, pelvic exercises lead to increased blood flow to the pelvic floor and aid in the rapid healing and revascularization of damaged cells and tissues.

A systematic review by Radzimińska et al. [46] attempted to identify the impact of PFMT on the QoL of women with UI among 2394 participants in 24 selected studies. After concluding the treatment, most patients in the experimental groups noted a significant improvement in their QoL. They demonstrated that PFMT is an effective treatment for UI that significantly improves the QoL of women with UI.

On the other hand, one study included in our review used electromyographic biofeedback as a tool to achieve muscle proprioception and muscle reeducation and coordination, eliminating inappropriate contraction patterns. Hagen et al. [29] published in their conclusions that no evidence was found on the severity of UI between PFMT plus electromyographic biofeedback and PFMT alone. The routine use of electromyographic biofeedback with PFMT should not be recommended.

In contrast to the previous study, Alouini et al. [47] concluded in a systematic review that PFMT alone or with biofeedback and electrostimulation was effective in reducing urinary incontinence and improving pelvic floor muscle contractions. PFMT, when compared with other interventions such as biofeedback, did not show significant differences but was superior to the control group. Although biofeedback therapy is not recommended on a daily basis, it is a useful tool to objectively measure the pressure exerted on the pelvic floor muscles, allowing patients to gain an estimate of the exertion force needed to obtain favorable results after training.

Modern advances in technology have favored the development of digital applications that can assist in the control of patients with UI and potentially benefit people who have limited access to healthcare facilities. Hou et al. [48] conducted a systematic review where they evaluated the effects of m-health app-based PFMT, QoL of users, and the patient’s global impression of improvement on the adherence to PFMT. They concluded that m-health app-based PFMT showed promising outcomes in exercise adherence. In a similar study, Åström et al. [49] concluded that women that turned to m-health for UI self-management advice had a reduced QoL and greater adherence to the EMSP compared to women that sought help in primary care. M-health might have reached a new group of women in need of UI treatment, proving that technological tools are having a good impact on the population, especially on those who find it difficult to access health services.

However, some women may still fail to comply with this due to their limited access to healthcare facilities and the lack of social and family support. In contrast, some women may remain unaware of these problems and their treatment owing to their lack of knowledge and education. Although most studies showed an improvement in the QoL, these results should be interpreted with caution because of the methodological limitations of some studies.

One limitation of our study was the heterogeneity in the designs of the studies. The included studies had a variety of intervention methods, measurement tools, and settings that restricted the subgroup analyses. Since the PFMT programs differed in their training durations, interpretations of the results should be made with caution. The main limitations of our study were the heterogeneity of the PFMT protocols included in the articles, as well as the durations of the exercises, since they varied considerably between the included articles. Additional limitations of our study were the methods and questionnaires used to identify the quality of life in the different studies. We could not obtain studies with similar sample sizes, and other articles we found were still in the protocol phase or had limited access. Finally, we did not have enough staff to sort and sift through the collected records.

## 5. Conclusions

Based on the obtained data, we can be certain that PFMT can improve the symptoms of SUI and other types of UI and reduce the number of involuntary leakages and symptoms in UI-specific symptom questionnaires. In addition, the review findings suggest that PFMT could be included in first-line conservative treatment programs for women with UI. However, the long-term efficacy and cost-effectiveness of PFMT need to be further investigated. In terms of UI severity, our results showed a significant improvement in the QoL of patients with mild UI. On the other hand, patients with severe UI showed minimal improvement in their QoL due to a greater number of comorbidities and even early stages of pelvic organ prolapse.

The treatment of UI (especially stress incontinence and mixed incontinence) through perineal reeducation positively contributes to improving the symptoms, strengthening the pelvic floor muscles, increasing the quality of life, and reducing the number of incontinence episodes. In addition, perineal reeducation can be considered a successful, noninvasive treatment that does not cause discomfort in the patient, constituting a good treatment option for men and women and various types of incontinence. The effectiveness of the results of perineal reeducation is related to compliance with the exercises and is independent from age. Training must be accomplished with a progressive difficulty level, perseverance, and motivation on the patient’s behalf. However, a larger number of studies would be necessary to specifically determine the type of exercise and the duration that is most appropriate for each type of incontinence.

## 6. Additional Comments

This systematic review and meta-analysis showed that more research should be carried out on this population. The lack of relevant and updated studies leads to insufficient information on this pathology, hindering possible diagnoses and follow-ups, especially in the elderly population. Finally, alternative ways of maximizing the effects of PFMT should be investigated.

## 7. Implications Section

These results built on the existing evidence of similar studies that showed statistically significant benefits in patients undergoing PFMT protocols, thus improving their QoL. Additionally, these findings could be taken as the cornerstone for the development of updated clinical practice guidelines that include PFMT in patients with UI, and this conservative treatment could be particularly beneficial in low- and middle-income countries due to its minimal economical investment, cost-effectiveness, and use as a preventive measure for UI.

While previous research has focused on a single type of strengthening exercise, our results showed that exercises such as yoga or Pilates can be adjusted to strengthen the pelvic floor and would be beneficial for women who have had previous experience in these types of exercises.

## Figures and Tables

**Figure 1 medicina-59-01004-f001:**
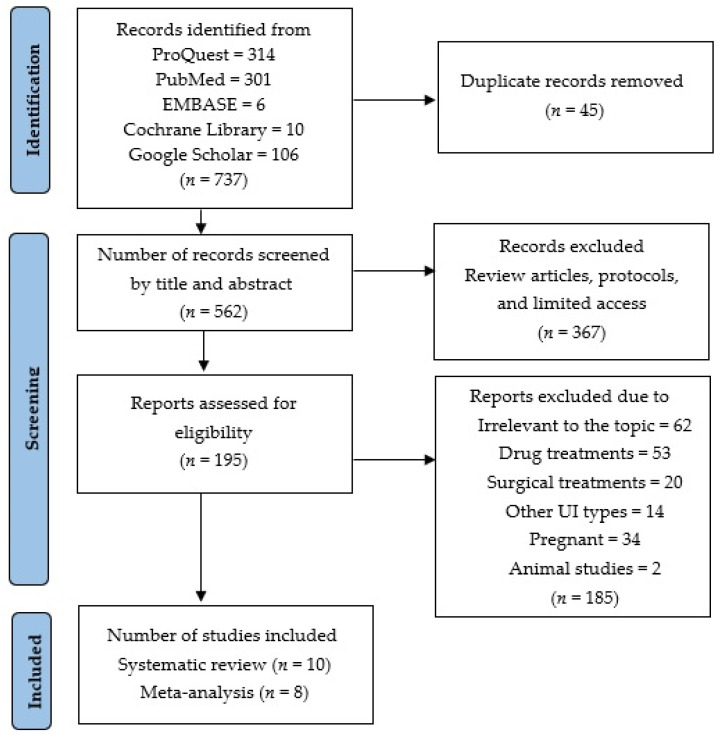
PRISMA table of systematic reviews and meta-analysis.

**Figure 2 medicina-59-01004-f002:**
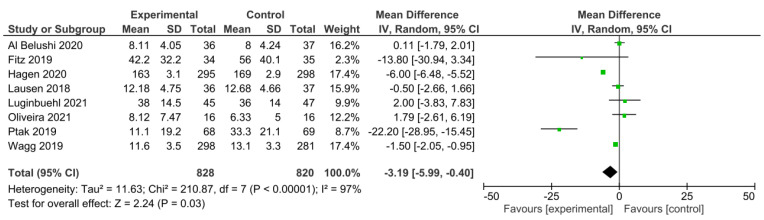
Forest plot of the comparison of the ICIQ-UI SF results after the intervention between the experimental and control groups [27,28,29,30,32,33,34,35]. Green boxes represent the mean difference of each study and the black diamond figure represents the overall effect estimate.

**Figure 3 medicina-59-01004-f003:**
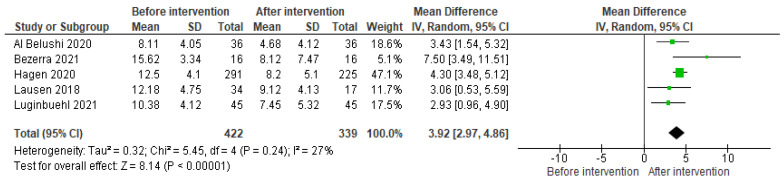
Forest plot comparing measurements of the ICIQ-UI SF before and after the intervention in the experimental group [27,28,29,30,35]. Green boxes represent the mean difference of each study and the black diamond figure represents the overall effect estimate.

**Figure 4 medicina-59-01004-f004:**
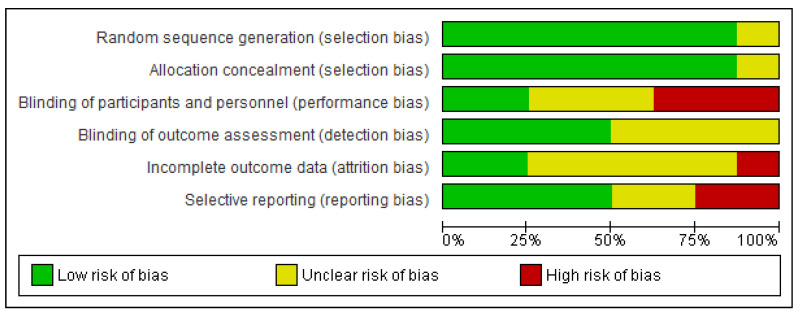
Risk of bias graph: review authors’ judgements about each risk of bias item presented as percentages across all included studies.

**Figure 5 medicina-59-01004-f005:**
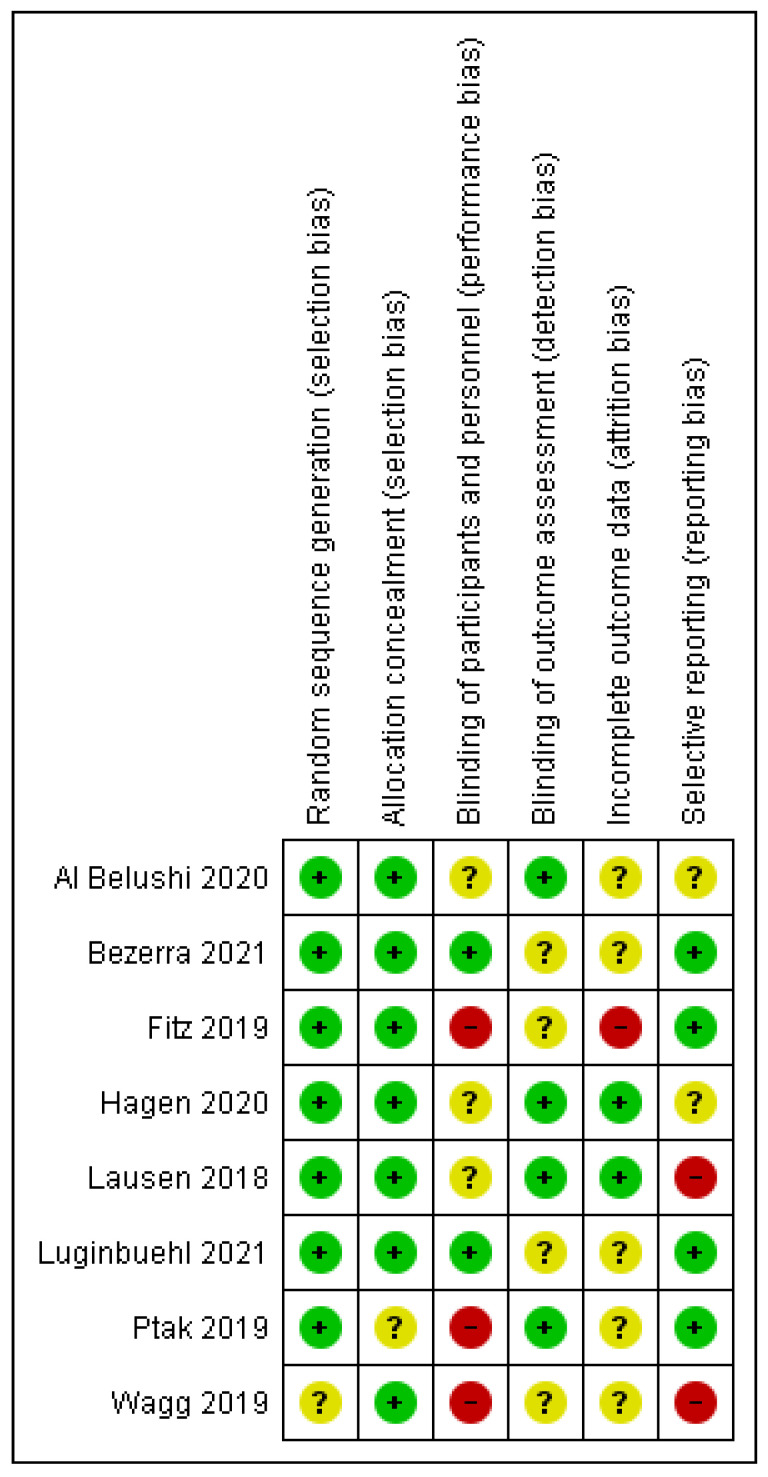
Risk of bias summary: review authors’ judgements about each risk of bias item for each included study [27,28,29,30,32,33,34,35].

**Table 1 medicina-59-01004-t001:** Studies obtained for the systematic review and meta-analysis.

N°	Author (s)	Journal/Publication	Title	Year	Place
1	Shafaq, S et al. [26].	Pakistan Armed Forces Medical Journal	Effects of Iyengar yoga on pelvic floor muscle strength and endurance among young females with SUI.	2022	Pakistan
2	Luginbuehl, H et al. [27].	International Urogynecology Journal	Involuntary reflexive PFMT in addition to standard training versus standard training alone for women with SUI.	2021	Switzerland
3	Bezerra, L et al. [28].	Games for Health Journal	Impact of PFMT isolated and associated with game therapy on MUI.	2021	Brazil
4	Hagen, S et al. [29].	BMJ	Effectiveness of PFMT with and without electromyographic biofeedback for urinary incontinence in women	2020	Scotland and England
5	Al Belushi, Z et al. [30].	Neuro-urology and Urodynamics	Effects of home-based PFMT on decreasing symptoms of stress urinary incontinence and improving the quality of life of urban adult Omani women	2020	Oman
6	Szatmári, E et al. [31].	Timisoara Physical Education and Rehabilitation	Efficacy of PFMT in improving symptoms of urinary incontinence and health related quality of life among parous and nulliparous women	2020	Romania
7	Fitz, F et al. [32].	International Urogynecology Journal	PFMT for female stress urinary incontinence	2019	Brazil
8	Ptak, M et al. [33].	BioMed Research International	The effect of PFMT on quality of life in women with SUI and its relationship with vaginal deliveries	2019	Poland
9	Wagg, A et al. [34].	Lancet Global Health	Exercise intervention in the management of urinary incontinence in older women in villages in Bangladesh	2019	Bangladesh
10	Lausen, A et al. [35].	BMC Women’s Health	Modified Pilates as an adjunct to standard physiotherapy care for urinary incontinence	2018	England

Abbreviations: SUI, Stress Urinary Incontinence; PFMT, Pelvic Floor Muscle Training; MUI, Mixed Urinary Incontinence.

**Table 2 medicina-59-01004-t002:** Characteristics of included studies.

Reference	Year	Country	Study Type	Number of Participants	Training Protocol	Outcome Measures	Main Findings
Shafaq, S et al. [26].	2022	Pakistan	IS	44	Two days of supervised group yoga courses and one day of home yoga conducted once a week for 8 weeks.	ICIQ-UI SF	Improving PMS and endurance, decreasing UI symptoms and distress hence improving the QoL.
Luginbuehl, H et al. [27].	2021	Switzerland	RCT	92	Involuntary reflexive PFMT triggered by whole-body movements such as jumps, for 16 weeks.	ICIQ-UI SF ICIQ-LUTS QoL	Score decreased about 3 points with no group differences at any point in time.
Bezerra, L et al. [28].	2021	Brazil	RCT	32	PFMT+GT group, interventions occurred twice a week for 8 weeks.	Manometry, Pad test, ICIQ-UI SF, PGI-I	Both treatments proved to be effective. All women reported being ‘‘much better or better’’
Hagen, S et al. [29].	2020	Scotland and England	RCT	593	Biofeedback PFMT, were given six appointments with a continence therapist over 16 weeks.	ICIQ-UI SF Oxford classification	Routine use of electromyographic biofeedback with PFMT should not be recommended.
Al Belushi, Z et al. [30].	2020	Oman	RCT	73	Unsupervised PFMT or a lecture with no PFMT for 12 weeks.	ICIQ-UI SF, MOGS, Manometry	PFMT is an effective treatment in reducing the severity of symptoms and improving the QoL.
Szatmári, E et al. [31].	2020	Romania	IS	42	PFMT among parous and nulliparous women for 10 weeks.	ICIQ-UI SF, KHQ	Parous women were 2 times more likely to report UI. They also had more severe symptoms and lower QoL.
Fitz, F et al. [32].	2019	Brazil	RCT	69	Outpatient PFMT and home PFMT under the guidance of a physiotherapist twice a week for 12 weeks.	Pad test, MOGS, I-QoL,	Both groups were satisfied after the treatment, even though this difference was not statistically significant.
Ptak, M et al. [33].	2019	Poland	RCT	137	Combined training of the PFMT and the transversus abdominis muscle was executed for 12 weeks.	ICIQ-LUTS QoL	Both exercises improve the QoL of women with SUI.
Wagg, A et al. [34].	2019	Bangladesh	RCT	579	PFMT was held twice a week for 12 weeks, with home exercises between classes.	EQ5D, CES-D-10,	Showed improvement in both intervention groups in the QoL and depression scales.
Lausen, A et al. [35].	2018	England	RCT	73	Modified Pilates classes as an adjunct therapy to standard physiotherapy care for UI, for 6 weeks.	I-QOL, ICIQ-UI SF, ICIQ-LUTS QoL, RSE	Improved self-esteem, decreased social embarrassment and lower impact on normal daily activities.

Abbreviations: IS, Intervention Study; RCT, Randomized Clinical Trial; PFMT, Pelvic Floor Muscle Training; PMS, Pelvic Muscle Strength; UI, Urinary Incontinence; QoL, Quality of Life; ICIQ-UI SF, International Consultation on Incontinence Questionnaire-Urinary Incontinence Short Form; ICIQ-LUTSQoL, ICIQ Lower Urinary Tract Symptoms Quality of Life Module; GT, Game Therapy; PGI-I, Patients Global Impression of Improvement for Incontinence; MOGS, Modified Oxford Grading System; KHQ, Kings Heath Questionnaire; I-QoL, Incontinence Quality of Life Questionnaire; EQ5D, five-dimensional EuroQoL Questionnaire; CES-D-10, Centre for Epidemiologic Studies Depression Scale; RSE, Rosenberg Self-Esteem Scale.

## Data Availability

Not applicable.

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
