# Peer review of "Effectiveness of Pelvic Floor Muscle Training on Quality of Life in Women with Urinary Incontinence: A Systematic Review and Meta-Analysis"

_medicina, 2023, doi:10.3390/medicina59061004_

Round 1

Reviewer 1 Report

The research content of the manuscript is relatively common, but the authors have made a systematic review for acceptable publication, however, please provide the registration numbers of review and Meta-analysis.

Acceptance is not recommended if the registration is not made before this work begins

Reviewer 2 Report

Effectiveness of pelvic floor muscle training on quality of life in women with urinary incontinence: A systematic review and meta-analysis

This is an interesting and important contribution to assess the effectiveness of pelvic floor muscle training on QoL of women with UI. The article is well written and is methodologically sound. Still, I believe some minor changes would improve its overall quality:

1.     Abstract: please include keyword used in the search, as well as the number of articles that were selected.

2.     Reference 2: However, women are most affected than men [2]. -please provide an explanation.

3.     Please provide a brief explanation of what PFM strengthening exercises are.

4.     Table 2: I would suggest authors to include a column with the number of participants.

5.     Please further discuss the limitations of this study.

6.     Please include an implications section.

Best wishes.

Reviewer 3 Report

The authors describe the effectiveness of PFMT on the QoL of women with urinary incontinence. It has become evident that PFMT is the first-line treatment in all women with urinary incontinence. This research would not add anything new to this subject; however, meta-analysis and systematic review of this work might make it persuasive to publish.
Overall the paper has been well-written, and the methodology and references are acceptable.In conclusion, please add your findings regarding the effectiveness of PFMT on QoL of different types of UI in terms of severity (does it improve women's QoL more in mild UI or severe UI?)

Line 82: IUM should change to MUI. 

Line 347: In this systematic review and meta-analysis, ten studies were used; they showed the 347

Line 349: of UI has been increasing over the years and that there was a

Line 379: and smoking were identified

Line 379: by Ptak et al. [33]. They (add punctuation before They)

Line 385: studied 233 women with urinary incontinence and their QoL, as well as their PFM strength; on their results, they published that

Line 389: spontaneous vaginal delivery

Line 404: On their results, they

Reviewer 4 Report

I would like to congratulate the authors on a well written manuscript.

In this study, the authors preformed a meta-analysis, identifying 10 studies analyzing the effects of pelvic floor muscle exercises on urinary incontinence.

The authors acknowledged their limitation of heterogeneity in the design of the different studies analyzed.

Although this has been extensively reported, the authors performed a thorough review which I found very interesting.

Round 2

Reviewer 1 Report

I reviewed the revised manuscript, along with the comments of the four reviewers and the responses of the authors, and I agree that the manuscript is accepted for publication in its current state.